# Multiplex Analysis of CircRNAs from Plasma Extracellular Vesicle-Enriched Samples for the Detection of Early-Stage Non-Small Cell Lung Cancer

**DOI:** 10.3390/pharmaceutics14102034

**Published:** 2022-09-24

**Authors:** Carlos Pedraz-Valdunciel, Stavros Giannoukakos, Ana Giménez-Capitán, Diogo Fortunato, Martyna Filipska, Jordi Bertran-Alamillo, Jillian W. P. Bracht, Ana Drozdowskyj, Joselyn Valarezo, Natasa Zarovni, Alberto Fernández-Hilario, Michael Hackenberg, Andrés Aguilar-Hernández, Miguel Ángel Molina-Vila, Rafael Rosell

**Affiliations:** 1Department of Cancer Biology and Precision Medicine, Germans Trias I Pujol Research Institute (IGTP), Campus Can Ruti, 08916 Badalona, Spain; 2Department of Biochemistry, Molecular Biology and Biomedicine, Autonomous University of Barcelona, Campus de Bellaterra, 08193 Barcelona, Spain; 3Laboratory of Oncology, Pangaea Oncology, Dexeus University Hospital, 08028 Barcelona, Spain; 4Department of Genetics, Facultad de Ciencias, Campus Fuentenueva s/n, Universidad de Granada, 18071 Granada, Spain; 5Exosomics SpA, 53100 Siena, Italy; 6B Cell Biology Group, Hospital del Mar Biomedical Research Park (IMIM), Barcelona Biomedical Research Park (PRBB), 08003 Barcelona, Spain; 7Vesicle Observation Centre, Laboratory of Experimental Clinical Chemistry, Department of Clinical Chemistry, Amsterdam UMC location University of Amsterdam, 1105AZ Amsterdam, The Netherlands; 8Cancer Center Amsterdam, Imaging and Biomarkers, 1105AZ Amsterdam, The Netherlands; 9Oncology Institute Dr. Rosell (IOR), Dexeus University Institute, 08028 Barcelona, Spain; 10Department of Computer Science and Artificial Intelligence, DaSCI., University of Granada, 18071 Granada, Spain; 11Catalan Institute of Oncology, Campus Can Ruti, 08916 Badalona, Spain

**Keywords:** circRNAs, extracellular vesicles, nCounter, lung cancer, NSCLC, liquid biopsies

## Abstract

Background: The analysis of liquid biopsies brings new opportunities in the precision oncology field. Under this context, extracellular vesicle circular RNAs (EV-circRNAs) have gained interest as biomarkers for lung cancer (LC) detection. However, standardized and robust protocols need to be developed to boost their potential in the clinical setting. Although nCounter has been used for the analysis of other liquid biopsy substrates and biomarkers, it has never been employed for EV-circRNA analysis of LC patients. Methods: EVs were isolated from early-stage LC patients (*n* = 36) and controls (*n* = 30). Different volumes of plasma, together with different number of pre-amplification cycles, were tested to reach the best nCounter outcome. Differential expression analysis of circRNAs was performed, along with the testing of different machine learning (ML) methods for the development of a prognostic signature for LC. Results: A combination of 500 μL of plasma input with 10 cycles of pre-amplification was selected for the rest of the study. Eight circRNAs were found upregulated in LC. Further ML analysis selected a 10-circRNA signature able to discriminate LC from controls with AUC ROC of 0.86. Conclusions: This study validates the use of the nCounter platform for multiplexed EV-circRNA expression studies in LC patient samples, allowing the development of prognostic signatures.

## 1. Introduction

With 350 deaths per day projected for 2022, lung cancer stands as the main cause of cancer-related mortality, leading the second highest incidence in the United States and Europe [1,2]. Treatments have proved to be more effective at the early stage of the disease, when lung cancer patients benefit from a significantly improved overall survival (OS) [3]. However, most cases are diagnosed at an advanced stage, with a 5-year survival rate dropping to only 5% in stage IV.

In order to achieve early detection, many challenges need to first be faced. Classical biopsy techniques for sampling and profiling of suspicious pulmonary nodules often involve invasive procedures. Limitations of such practices include restricted access to the nodules, which regularly compromise the quality and quantity of extracted biopsy specimens. Heterogeneity of resected samples also hampers the use of these methods, especially for tumor identification [4].

Liquid biopsies offer a minimally invasive procedure for sampling, providing a practical tool for continuous monitoring of lung cancer patients [5], being also actively investigated for early detection [6]. Despite the slow progression on the development of liquid biopsies in this area, many possible biomarkers have been proposed in the last few years, including circulating tumor DNA (ctDNA), cell-free RNA (cfRNA), circulating tumor cells (CTCs), proteins, extracellular vesicles (EVs) and tumor educated platelets (TEPs).

Lung cancer elicits massive changes in RNA metabolism, reflecting both in the tumor transcriptome and in the circulating EV and TEP cargo. EVs contain different RNA molecules, including mRNA and non-coding RNAs such as miRNA or circular RNAs (circRNAs) [7,8]. The circRNA transcripts are generated by post-transcriptional circularization of the 5′ and 3’ends in an alternative process called back-splicing. Their circular structure makes most of them resistant to exonucleases and, therefore, robustly stable RNA molecules, compared to the canonical (linear) mRNA. CircRNAs seem to play an important role in human homeostasis [9,10]. Moreover, it has been reported that aberrant expression of certain circRNAs can promote cancer development and progression [11]. Additionally, some circRNAs have been investigated as liquid biopsy biomarkers for the early detection of lung cancer and other solid tumors [12,13]. However, the lack of consensus on a robust and standardized protocol for circRNA quantification is holding back the development of clinically applicable assays.

RT-qPCR, microarrays and RNAseq are the three methods most commonly used in circRNA research. However, the RT-qPCR does not allow high-throughput analysis; microarrays have a limited dynamic range of RNA detection; and RNAseq is associated with high cost, long time-consuming protocols, and high grade of complexity when it comes to data analysis.

An alternative technique for multiplex analysis of circRNA is nCounter, which provides a cost-effective automated solution for analysis of more than 800 targets with minimal hands-on time, providing highly reproducible data in less than 48 h. nCounter is based on the detection of RNA of interest using target-specific probe pairs. Each pair comprises of a reporter probe with a unique color combination at the 5’- end, allowing specific recognition of the gene of interest; and a capture probe carrying a molecule of biotin, which provides a molecular grip to the nCounter cartridge, allowing downstream digital detection [14]. The expression of a particular gene is then calculated by counting the number of times a specific color-coded probe is detected. This technology has been embraced in translational research, including the development and validation of liquid biopsies, due to its capability of working with a low quantity of highly degraded samples [15,16]. Recent studies reported the use of nCounter for the study of several categories of circulating biomarkers [17,18,19,20,21], including EV-derived DNA [22], miRNA [23,24], mRNA [25] and circRNA [26]. However, nCounter analysis of EV-circRNAs has not been investigated for early detection of lung cancer. Here, we report the development of a protocol for EV enrichment from plasma followed by RNA purification and circRNA analysis by nCounter. 

Then, we analyzed liquid biopsies from non-cancer donors and early-stage non-small cell lung cancer (NSCLC) patients and applied machine learning (ML) to develop a prognostic signature.

## 2. Materials and Methods

### 2.1. Patient Samples

The study was carried out in accordance with the principles of the Declaration of Helsinki, under an approved protocol of the institutional review board of Quirón Hospitals. We obtained and documented written informed consent from all the patients. A total of 36 samples from early-stage NSCLC (stages IA to IIIA) were selected from our institution, along with 30 samples from non-cancer controls (Table 1). Clinical information from patients and controls included age, gender, smoking status, tumor histology and stage, when applicable. All samples were de-identified before further processing for confidentiality purposes.

### 2.2. Plasma Processing

Around 10 mL of whole blood was collected from the participants enrolled in the study using sterile EDTA Vacutainer tubes (BD, Plymouth, UK) and processed within the next 2 h. Blood samples were centrifuged twice at 2000× *g* at room temperature (RT) in a Rotina 380 R centrifuge (Hettich, Tuttlingen, Germany) for 10 min to separate plasma from red/white blood cells, platelets, and cell debris. Aliquoted plasma samples were then stored at −80 °C until downstream processing.

### 2.3. Enrichment of EVs

EVs were isolated from plasma using differential ultracentrifugation (UC) as described previously [27] or the miRCURY Exosome Serum/Plasma Kit (Qiagen, Hilden, Germany).

In the case of UC, 500 μL plasma samples were transferred into 15 mL sterile high-speed centrifuge tubes (VWR-Avantor, Philadelphia, PA, USA), filled up with sterile 1× phosphate-buffered saline (PBS) and centrifuged twice at 10,000× *g* for 30 min at 4 °C in a Sorvall RC 6 Plus centrifuge (Thermo Fisher Scientific, Waltham, MA, USA). Supernatants were then transferred into UC tubes (Beckman Coulter, Brea, CA, USA), equilibrated with sterile 1× PBS, and spun twice at 70,000× *g* for 1 h at 4 °C in the Sorvall WX Ultra 100 centrifuge (Thermo Fisher Scientific). The EV enriched pellets were resuspended in 100 μL sterile PBS and stored at −80 °C until used. EV enrichment with the miRCURY Kit was performed as described [25]. Debris was cleared from 500 μL plasma samples with thrombin and subsequent centrifugation at 10,000× *g* for 5 min at RT. Samples were then incubated with Precipitation Buffer overnight at 4 °C and centrifuged twice (500× *g*, 5 min at RT). Supernatants were discarded, EV enriched pellets were resuspended in 270 μL of Resuspension Buffer and stored at −80 °C until used.

### 2.4. Transmission Electron Microscopy (TEM)

Visualization of EV samples was performed by the TEM service of the Universitat Autónoma de Barcelona (UAB). A volume of 3.9 µL of EV-enriched sample was blotted onto a Holey Carbon Film Supported Nickel Grid (Merck, Darmstadt, Germany) previously glow-discharged in a PELCO easiGlow glow cleaning system (Ted Pella Inc, Redding, CA, USA). Next, the grid containing the sample was plunged into a Leica EM GP cryo-work station (Leica, Wetzlar, Germany) comprising a liquid ethane bath cooled to −180 °C, and subsequently transferred and visualized in a JEOL 2011 TEM (Jeol Ltd., Tokyo, Japan) operating at 200 kV. Samples were kept at −180 °C during the observation and captures were obtained with a Gatan Model 895 UltraScan 4000 4k × 4k CCD camera (Gatan Inc, Pleasanton, CA, USA). Image processing was performed using ImageJ software (version 1.8.0, National Institutes of Health, Bethesda, MD, USA).

### 2.5. Nano-Flow Cytometry Measurements

The volume of EV samples was brought to 500 µL with sterile PBS. Size-exclusion chromatography (SEC) columns (qEVoriginal/35 nm, Izon Science, Oxford, UK) were equilibrated with 20–30 mL of sterile PBS and eluted using the same buffer. Collection started immediately after loading the sample into the column, according to manufacturer instructions. Eluted EV-enriched samples were directly analyzed with the nanoFCM (NanoFCM Ltd., Nottingham, UK), a nanoparticle flow cytometer. Instrument calibration with standard beads enabled accurate measurements of both size and concentration of 40–200 nm particles through the detection of their side scatter [28].

### 2.6. RNA Isolation and DNase Treatment

EV-enriched samples were treated with 4 μg/mL of RNase A (Sigma-Aldrich, Burlington, MA, USA) for 1 h at 37 °C, to eliminate any non-vesicular RNA. TRI Reagent (MRC, Cincinnati, OH, USA) was added to a final volume of 1 mL and incubated at RT for 20 min. Then, 200 μL of a Chloroform and Isoamyl Alcohol dilution (24:1) (Panreac Química SLU, Barcelona, Spain) were added, followed by vigorous shaking and centrifugation at 12,000× *g* for 15 min at 4 °C. Upper fraction was collected, and RNA was precipitated by adding 2.5 μL of glycogen (Merck) and 500 μL 2-propanol (Merck), followed by incubation at RT for 10 min and further centrifugation at 12,000× *g* for 10 min at 4 °C. RNA pellet was then washed with 75% ethanol, dried at 95 °C for 3 min and resuspended in 12 μL of nuclease-free water.

The DNA-free DNA Removal Kit (Thermo Fisher Scientific) was used to eliminate any DNA remaining in the samples. Following the manufacturer’s protocol, 0.75 μL of DNase buffer and 1 μL enzyme were added to 7.5 μL RNA sample and incubated at 37 °C for 30 min. A volume of 0.75 μL of DNase inactivation reagent was then added to the reaction, incubated for 2 min at RT and centrifuged for 1.5 min at 10,000× *g* and RT. The supernatant containing EV-RNA was then transferred to a fresh tube and stored at −80 °C until further use.

### 2.7. RT-qPCR and Sanger Sequencing Analysis

RT-qPCR and Sanger sequencing of circRNA junction sites were performed as previously described [29]. Divergent primers and probe sets were designed using Primer Express 3.0 Software (version 3.0.1, Applied Biosystems, Waltham, MA, USA) with the probes spanning the circRNA junction site (Table 2). Five microliters of EV-RNA was converted into cDNA using the M-MLV reverse transcriptase enzyme and random hexamers (both from Invitrogen, Waltham, MA, USA). A 1:3 dilution of cDNA was performed, and 2.5 µL were added to the Taqman Universal Master Mix (Applied Biosystems) in a 12.5 µL reaction containing a specific pair of primers and probe for each circRNA. Three replicas of each sample were run for the quantification of the expression of each assessed circular transcript. Quantification of gene expression was performed using the QuantStudioTM 6 Flex System (Applied Biosystems) and the comparative Ct method.

For Sanger sequencing, 10 µL of each PCR product was subjected to electrophoresis in a 2× agarose gel (100 V, 30 min) and visualized under UV light (E-Gel™ Safe Imager™ Real-Time Transilluminator, Invitrogen) after electrophoresis (E-Gel™ iBase™ Power System, Invitrogen). Five microliters of each cDNA sample were purified using the PCR ExoSAP-IT Product Clean up Reagent (Applied Biosystems). Sequencing PCRs were set up using the BigDye Terminator v3.1 Cycle Sequencing Kit (Applied Biosystems), forward primer, cDNA and water in a final volume of 20 µL and performed using a Verity 96-well thermal cycler (Applied Biosystems). After sequencing amplification, samples were loaded into a 96-well plate and subjected to Sanger sequencing using the 3130 Genetic Analyzer (Applied Biosystems).

### 2.8. nCounter Processing

The nCounter Low RNA Input Amplification Kit (NanoString Technologies, Seattle, WA, USA) was used to retrotranscribe and pre-amplify 4 μL of EV-derived RNA in a Verity thermal cycler (Applied Biosystems) following NanoString’s guidelines. Briefly, samples were denatured at 95 °C for 10 min and hybridized for 18 h at 67 °C. Our custom-made nCounter panel (including 78 circRNAs, 6 linear reference genes and 4 mRNAs [30] was used to analyze EV-derived pre-amplified cDNA according to the manufacturer’s instructions. RCC files containing data outputted by the NanoString nCounter Flex System (NanoString Technologies) from each run were exported to the nSolver Analysis Software (Version 4.0.70, NanoString Technologies).

### 2.9. Differential Expression Analysis

Raw count nCounter values were exported to Microsoft Excel (Version 16.40, Microsoft, Redmond, WA, USA) using nSolver Analysis Software. The background was calculated for each sample as (geo)mean ± 2SD of the negative probe counts (NCs) Raw counts lower than the background were automatically excluded from further analysis. The raw circRNA counts were normalized using the total number of counts of the sample and multiplied by 10,000. Differential expression analysis was performed comparing the means of the normalized counts for each circRNA in the early-stage NSCLC vs. non-cancer controls. The circRNAs with a fold change >1 and *p*-value < 0.05 were considered as differentially expressed (DE).

### 2.10. Data Pre-Processing and Normalization for Signature Development

Raw RCC-formatted data files were exported from the nSolver Analysis Software (NanoString Technologies). R (Version 4.0.3, R Core Team and the R Foundation for Statistical Computing, Vienna, Austria) and R studio (Version 2021.09.0, RStudio PBC, Boston, MA, USA) were used for pre-processing and normalization analysis of the imported files. Initial evaluation of the quality and integrity of the RCC data was performed using the NanoStringQCPro (Version 1.22.0) package. During this process, we looked for potential outliers based on the performance of standard control metrics provided by NanoString, such as Imaging, Binding Density, Positive Control Linearity, and Limit of Detection. After this first pre-analytical step, samples were subjected to supplementary exploratory examination, including Principal Component Analysis (PCA) and interquartile range (1.5 IQR rule) analysis. Samples found as outliers by both methods were then excluded from downstream analyses.

NCs were employed to exclude lowly expressed circRNAs with excessive background noise. The arithmetic mean of the NC ± 2SD was subtracted from each endogenous circRNA for each sample. Any transcript scoring a value below 0 in more than 75% of the analyzed samples was then excluded from further analysis. PCA plot was used to re-assess the data after the aforementioned filtering step. Technical variability correction and normalization were performed using the RUVSeq/RUVg function (Version 1.24.0) and DESeq2 (Version 1.30.1) packages (RUVseq-DESeq2). First, the RUVg function was used to estimate the unwanted variation among samples based on the DE genes. DESeq2 and edgeR (Version 3.32.1) performed a first pass DE analysis and the intersected least significant genes (with adjusted *p*-value above 0.1) were used as “in silico empirical” negative controls. DESeq2 was then utilized with default parameters along with the RUV factors to perform the normalization of the raw filtered data. The normalization performance was assessed using the standard relative log expression (RLE) plot.

### 2.11. Machine Learning (ML) for Signature Development

The Recursive Feature Elimination (RFE) algorithm along with leave-one-out cross-validation (LOOCV) and the random forest (RF) classifier were used to perform feature selection on the normalized data previously generated by RUVseq-DESeq2. The optimal number of features was automatically selected by keeping only those yielding best performance after cross-validation. These final features were to constitute the prognostic signature. To test the predictive power of the selected signature, extra trees classifier (ETC), k-nearest neighbor (KNN) and RF models were built using default parameters. The 5-fold cross validation (5-CV) algorithm was applied for this purpose. During this process, the dataset was randomly split into k-folds (k = 5), being 4/5 of the data used to train the model, while the remaining 1/5 was used to test its behavior. The classifier showing the highest area under the ROC curve (AUC ROC) value was selected as the final model. Signature scores for each sample were obtained from the final model. A confidence threshold of 0.5 was considered for the calculation of the positive and negative predictive values (PPV–NPV). Additional statistical indicators such as accuracy, sensitivity, specificity, and Cohen’s κ were also calculated.

### 2.12. Univariant and Multivariant Analyses

Association between clinical characteristics and ML-generated signature was assessed with a univariate Cox proportional-hazard regression model. Odds ratios, with a Confidence Interval (CI) of 95% was calculated using the MedCal Statistical Software (MedCalc Software Ltd. Odds ratio calculator. https://www.medcalc.org/calc/odds_ratio.php. Accessed last on 5 September 2022). Multivariant analysis using logistic regression was performed using SAS software (v9.4, SAS Institute, Cary, NC, USA). Significance was set at *p* < 0.05 for all statistical tests.

## 3. Results

### 3.1. Enrichment of Plasma EVs and Workflow Development for nCounter CircRNA Analysis

Two replicated 500 µL plasma samples from an early-stage NSCLC patient and a non-cancer control were submitted to EV enrichment by ultracentrifugation (UC) or using the miRCURY Exosome Serum/Plasma kit. Enriched EVs were characterized by transmission electron microscopy (TEM) and nanoparticle flow cytometry via nanoFCM. TEM images revealed different clusters of diverse-sized EVs (30 to 300 nm, all within the reported EV size range [31,32,33]) in all samples regardless of the enrichment method used (Figure 1a). Samples extracted using the miRCURY kit showed a higher proportion of vesicles with an exosome-like size range (30–100 nm) by TEM, compared to the more heterogeneous UC samples (Figure 1a). NanoFCM analysis revealed a higher concentration of 40–100 nm particles in samples enriched using the miRCURY kit (Figure 1b). In addition, nanoFCM indicated a higher number of particles/mL in the NSCLC patient sample when compared to the control, both in the UC and miRCURY preparations (Figure 1b).

Next, different volumes of plasma (500 µL, 1000 µL and 1500 µL) from a NSCLC patient were tested in duplicates to assess the effect of initial volumes on downstream circRNA analysis by nCounter using the custom panel we previously developed [30]. Since RNA concentration from EV enriched samples has been demonstrated to be insufficient for direct nCounter analysis [25], pre-amplification steps of 14 and 20 cycles were tested. The utmost total number of counts was achieved using an input of 500 µL both with 14 and 20 cycles (14,151 ± 1864 and 686,525 ± 345,655, respectively; Figure 2a). Consequently, 500 µL of plasma was also the volume allowing the detection of more circRNAs (*n* = 27.5 ± 4.95 and 33 ± 7.07 for 14 and 20 cycles, respectively; Figure 2b), even if only those with a score above 10 counts after background removal were selected (Appendix A, Appendix A).

Different amplification cycles (10, 12 and 14) were subsequently tested in a 500 µL plasma sample. The highest number of raw counts was obtained with 14 cycles (Figure 3a). Regarding the number of circRNAs, 10 and 12 cycles yielded similar results (*n* = 51.5 ± 9.19 and 52.5 ± 7.78 respectively). More circRNAs were detected at 14 cycles (*n* = 59 ± 16.97) with a high variability between replicates (Figure 3b, Appendix A). In view of these results, we selected for EV-circRNA analysis a protocol that included 500 µL of plasma input, EV enrichment with the miRCURY kit, extravesicular RNA elimination with RNase A, EV lysis and RNA extraction with TRI reagent, retrotranscription and nCounter analysis with a 10-cycle preamplification step (Figure 4).

The repeatability of the protocol was first tested by submitting to nCounter duplicates of a preamplified plasma sample. A strong correlation between the normalized counts was found between the duplicates, represented by a Pearson’s *r* of 0.98, *p* < 0.0001 (Figure 3c). When the same plasma sample was re-purified and re-analyzed, nCounter results also showed a strong correlation with the initial duplicates (Pearson’s *r* = 0.90–0.91; *p* < 0.0001) (Figure 3d).

### 3.2. CircRNA Expression in Plasma EV Samples

Plasma from 66 individuals, 36 early-stage NSCLC patients and 30 non-cancer donors, were analyzed using the protocol previously described in Section 3.1 (Figure 4). An average of 40 ± 14 EV-circRNAs per sample were detected in controls vs. 47 ± 9 in the NSCLC cohort. This difference was found to not be significant by the Mann–Whitney U test (Figure 5a). Among the 78 circRNAs included in the panel, 70 were detected in at least one NSCLC sample and 68 in at least one non-cancer control. A total of 66 EV-circRNAs were shared by both cohorts, while four EV-circRNAs were exclusive to NSCLC patients and two to non-cancer donors (Figure 5b, Appendix A).

DE analysis revealed eight circRNAs significantly upregulated in EV-enriched samples from NSCLC patients vs. controls; namely circular Erythrocyte Membrane protein Band 4.1 Like 2 (circEPB41L2), circular Core 1 Synthase, Glycoprotein-N-Acetylgalactosamine -3-Beta-Galactosyltransferase 1 (circC1GALT1), circular Zinc Finger RNA Binding Protein (circZFR), circular Ubiquitin Specific Peptidase 3 (circUSP3), circular Zinc Finger CCHC Domain-Containing Protein 6 (circZCCHC6), circular Cyclin B1 (circCCNB1), circular DENN Domain Containing 1B (circDENN1B) and circular Homeodomain Interacting Protein Kinase 3 (circHIPK3) (Figure 5c). Of them, only circZFR and circC1GALT1 showed <10 counts in each cohort (Appendix A). To validate these results, we tested the expression circZCCHC6 and circHIPK3 by RT-qPCR. Divergent primers and probes spanning the junction sites were designed for the specific amplification of these two circular transcripts (Table 2) in samples previously assessed by nCounter with sufficient remaining material. Gel electrophoresis of the RT-qPCR products revealed bands matching the size of expected amplicons and subsequent Sanger sequencing confirmed the expected junction sites in the circRNAs (Figure 6a,b). Among the six samples analyzed by RT-qPCR, four and six samples produced satisfactory results for circZCCHC6 and circHIPK3 respectively. A trend between nCounter counts and RT-qPCR ∆∆Cts was observed for both circRNAs (Figure 6c–d), with circZCCHC6 showing a strong correlation (Pearson’s *r* = 0.99; *p* = 0.0076) (Figure 6c).

### 3.3. Development of a CircRNA-Signature Associated with Early-Stage NSCLC

Interquartile range analysis classified 9/66 samples as potential outliers (Figure 7a) and PCA revealed that they deviated from the main cluster of observations (Appendix A). Consequently, these nine samples were excluded from further analysis.

Then, different R packages including DESeq2, edgeR, RUVSeq and their combination were tested in order to select the normalization approach that best adapts to our data. As a result, RLE plots indicated a superior performance of RUVSeq-DESeq2 versus the other combinations (Figure 7b and Appendix A). Consequently, RUVSeq-DESeq2 normalization was selected for the rest of the study.

Next, ML was performed using RFE along with RF classifier and LOOCV, as described in Methods, in order to obtain a signature associated with NSCLC. As a result, ETC was selected as the best model, with a signature of 10 circRNAs (including circular Family With Sequence Similarity 13 Member B -circFAM13B, circular ADAM Metallopeptidase Domain 22 -circADAM22, circular UBX Domain Protein 7 -circUBXN7, circZCCHC6, circular Integrin Subunit Alpha X -circITGAX, circular Retinol Dehydrogenase 11 -circRDH11, circEPB41L2, circular CDC Like Kinase 1 -circCLK1, circular Phenylalanyl-tRNA Synthetase Subunit Alpha -circFARSA, and circular Phosphoinositide-3-Kinase Regulatory Subunit 1 -circPIK3R1) showing an AUC ROC of 0.86 (Figure 8a). Signature scores were found to be statistically different when comparing early-stage NSCLC and non-cancer controls (Mann–Whitney U test, *p* < 0001; Figure 8b). The sensitivity and specificity of the ETC signature were of 90% (CI = 73.47–97.89%) and 81% (CI = 61.92%%–93.70%) respectively, outperforming the RF and KNN classifiers (Table 3). The accuracy achieved with ETC was 86%, resulting in 49 out of the 66 cases being correctly classified (Figure 8c).

Then, a univariate analysis was performed to explore the association of the ETC circRNA signature with gender, age, smoking, cancer status and tumor stage (Figure 9a). A statistically significant correlation was found for the signature with age (odds ratio = 24.91, *p* < 0.0001), and particularly cancer status (odds ratio of 39.6, *p* < 0.0001).

To further evaluate the implication of age and cancer status on the ML-developed signature, we first performed an exploratory study assessing the interconnexion of both variables by performing a chi-square test. As a result, a strong association between age and cancer status was found, with a *p* < 0.0001 (Table 4).

Next, a multivariate analysis was carried out. Results not only demonstrated dependency of these two variables, but also showed a statistically significant correlation between the signature and cancer status (*p* = 0.0036, Table 5, Figure 9b). No correlation was found between age and presented signature, in this regard (*p* = 0.0784, Table 5, Figure 9b)

## 4. Discussion

EVs are released by most cell types and play an important role in cancer cell communication. Many publications have demonstrated the role of EVs as key modulators in cancer progression [34,35], which requires intercellular communication mediated by the horizontal transferring of biological information via the EV cargo of proteins, DNA and coding/non-coding RNA, including circRNAs. Therefore, analysis of EVs can provide a snapshot of the tumor and serve as a valuable tool to discover liquid biopsy biomarkers. CircRNAs are highly enriched in EVs [7] and show a relatively high stability compared to other forms of RNA [8]. Several studies have highlighted their potential as liquid biopsy biomarkers [12] but current limitations in circRNA quantification methods are limiting their implementation in the clinical setting. Consequently, new, and robust protocols for circRNA analysis are needed. The nCounter platform has gained popularity among translational investigators for transcriptional research not only for solid biopsies but also for EV samples. However, studies focusing on circRNA analysis by nCounter are limited and mostly restricted to tissue specimens [30,36,37,38,39,40]. In particular, to the best of our knowledge, nCounter has never been applied to the analysis of circRNA in liquid biopsies of lung cancer patients. Consequently, we developed a comprehensive protocol for nCounter-based EV-circRNA expression analysis, from EV enrichment to differential expression and subsequent ML analysis. Key points in this protocol were the initial volume of plasma, the EV purification method and the number of cycles for the pre-amplification step prior to nCounter testing.

UC is currently still the method of choice for EV isolation in the research setting and we have previously demonstrated its utility for the downstream analysis of cell line-derived EV circRNAs [41]. However, ultracentrifuges are not usually available in clinical laboratories, while precipitation-based kits such as the miRCURY Exosome Serum/Plasma Kit represent an easily implementable option with a simple, on-the-bench protocol and short hands-on time. In our study, we compared the two methodologies using plasma samples from an NSCLC patient and a healthy donor. The presence of EV-like particles in all preparations was confirmed by TEM and nanoFCM. Interestingly, a more uniform EV population with an exosomal size-range was found by TEM in both cancer and control samples processed with the miRCURY kit, along with a higher concentration of 40–200 nm particles observed by nanoFCM. A possible explanation to this event could be a size-selective enrichment attributed to this type of precipitation-based preparations, as previously reported in serum samples [42]. This finding prompted us to select miRCURY for further assay development. In addition, a higher number of EV-like particles was observed in the cancer sample compared to the control, regardless the isolation method used. Although a higher number of samples should be analyzed for further confirmation, preliminary results are in agreement with previous reports indicating a higher abundance of EVs in cancer patients [43].

Finally, adding to the evidence provided by TEM and nanoFCM, a treatment with RNase A was applied to EV-enriched samples prior to EV lysis and incorporated into our protocol to eliminate any extravesicular RNA. The resulting and subsequently analyzed RNA proved to be protected from the digestion of cited ribonuclease, indicating a vesicular origin of the transcripts.

In a previous study, a volume of 500 µL of plasma was found to be sufficient for the analysis of EV-derived mRNA by nCounter [25]. Here, we compared several plasma volumes and found that 500 µL outperformed 1000 and 1500 µL for circRNAs analysis, both in terms of the number of circRNA molecules detected and total counts. A possible explanation for these results may rely on saturation issues with the circRNAs/reporter–probe complexes when a higher plasma input is applied, which impede a correct molecule identification by the digital analyzer. Regarding the number of cycles for the preamplification step, we investigated a range from 10 to 20 in an effort to reduce amplification-related background noise to a minimum, and we found that a 10-cycle pre-amplification step yielded adequate results.

Then, we applied our protocol to assess circular transcripts in early-stage NSCLC samples (*n* = 36) and to non-tumor controls (*n* = 30). We found that eight circRNAs were found differentially expressed between the two cohorts. Among them, circEPB41L2, circZCCHC6 and circHIPK3 showed the highest number of counts in early-stage cancer patients (Appendix A). Interestingly, we previously found circEPB41L2 differentially expressed in FFPE tissues of early-stage lung cancer patients [30] and found that it displayed four binding sites with hsa-miR-942, which has been described as an activator of the Wnt/β-catenin signaling pathway [44,45] in colorectal and esophageal cancers. Our results warrant further investigation in the biology of this circRNA to characterize its role in lung cancer. Regarding circHIPK3, it has been extensively investigated in lung cancer and found to exert a dual activity over miR-149 [46] and mir-124 [47,48], inducing cell proliferation and inhibiting apoptosis. Our results are in agreement with these findings, since circHIPK3 was upregulated in EV samples from early-stage NSCLC patients. Finally, circZCCHC6 has been recently described to regulate lysophosphatidylcholine acyltransferase 1 (LPCAT1) levels via miR-433-3p [49] in lung cancer. We used circinteractome (www.circinteractome.nia.nah.gov) to investigate possible additional miRNA binding sites, finding matches for 7 additional transcripts (miR-579-3p, miR-623, miR-1197, miR-1304 miR-548l, miR-605 and miR-935). All these miRNAs have been reported to be downregulated in lung tumors and have been related with poor prognosis, tumor growth and metastases [50,51,52,53,54,55,56].

ML and other computational methods based on artificial intelligence (AI) have emerged in the last decade for multileveled analysis of different datasets. In particular, ML enables computers to make predictions by finding patterns within analyzed data [57], offering a novel approach for the development of predictive signatures that often reach a higher predictive value than biomarkers found by differential expression analyses. Consequently, we decided to use ML in our study. To this end, we developed a pipeline with several steps. First, using IQR and PCA plots, we identified nine outliers, which were excluded from downstream analyses. An RLE plot from each different normalization procedure was generated, showing a higher performance of the RUVSeq-DESeq2 function when compared to the other combinations (Figure 7b and Appendix A). Finally, we used RFE along with LOOCV and the RF classifier as the feature selection algorithm to automatically determine the most significant circRNAs which are best suited for the construction of the prognostic signature. The final 10-circRNA signature included two of the eight circular transcripts previously found by differential expression analysis and eight additional transcripts, including circFARSA. Interestingly, circFARSA has been described as a plasma biomarker of NSCLC [58], promoting tumor invasion and metastases via the PTEN/PI3K/AKT axis [59].

Since we did not sort EV populations, we could not verify the vesicular cell or tissue origin of the circRNAs included in the ML signature nor the origin of the circular transcripts, either cancer cells or tumor microenvironment. Also, we did not investigate the biological role of the circRNAs, being out of the scope of our work.

In addition, while multivariate analysis could demonstrate that classification accuracy of presented signature is based on cancer status and no other clinicopathological characteristics (Figure 9), the lack of > 60-year-old individuals was a limitation in the study. The inclusion of equivalent cohorts in terms of age should be taking into consideration for the design of forthcoming validation studies.

Finally, all 36 cancer samples included in this study were lung adenocarcinomas, with the exception of 4 squamous carcinoma and 5 NSCLC samples with unknown histological subtype. A uniform inclusion of the different lung cancer histologies is suggested for future validation studies to assess the predictive power of the signature for other subtypes of NSCLC.

## 5. Conclusions

We have demonstrated the feasibility of using nCounter for the multiplex study of plasma-EV circRNAs in liquid biopsies of lung cancer patients, including differential expression analysis and development of predictive ML signatures. Further studies of larger cohorts are warranted in order to determine the clinical applicability of such signatures.

## Figures and Tables

**Figure 1 pharmaceutics-14-02034-f001:**
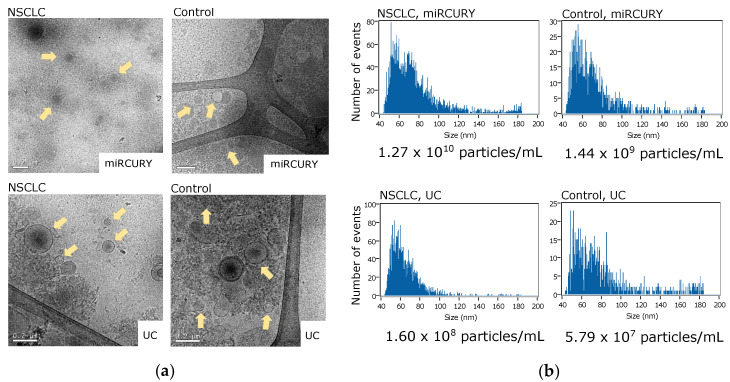
Characterization of extracellular vesicles (EVs) enriched either by differential ultracentrifugation (UC) or precipitation using the miRCURY Exosome Serum/Plasma kit. (**a**) Observation of EV samples on transmission electron microscopy (TEM). Yellow arrows point out EVs of different sizes. Scale bars indicate 200 nm.; (**b**) Nanoflow cytometry (nanoFCM) profiles of EV samples showing size and concentration of 40–200 nm particles.

**Figure 2 pharmaceutics-14-02034-f002:**
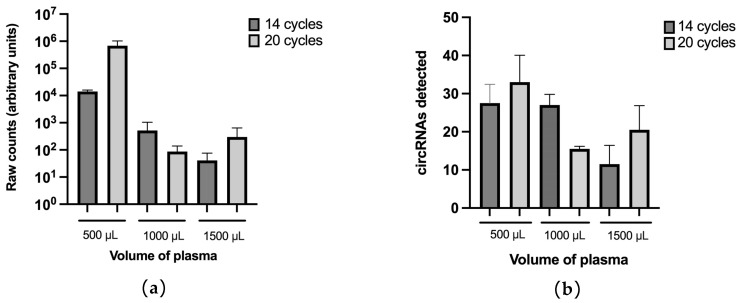
Plasma input testing. (**a**) Total number of counts and (**b**) number of circRNAs detected by nCounter with each of the volumes of plasma tested with 14 and 20 cycles of pre-amplification. Plasma from a NSCLC patient was used for this purpose. Error bars indicate standard deviation.

**Figure 3 pharmaceutics-14-02034-f003:**
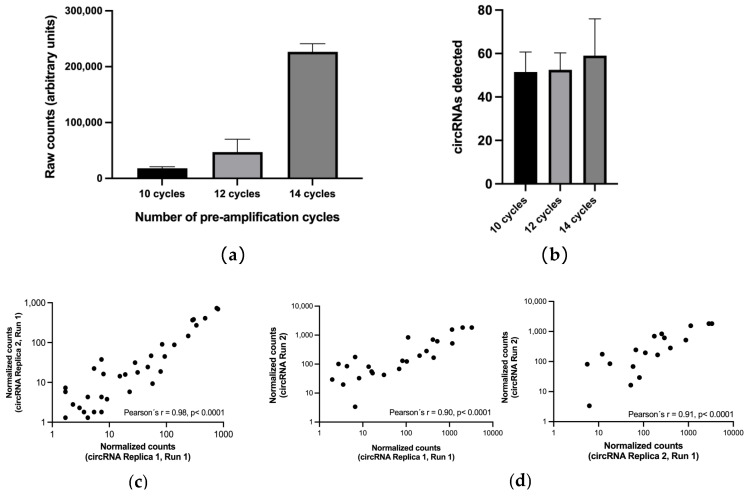
Testing of a different number of pre-amplification cycles. Effect of the number of pre-amplification cycles (10, 12 and 14) on (**a**) the total number of raw counts and (**b**) total number of circRNAs detected. Error bars indicate standard deviation; (**c**) Correlation of the two technical nCounter duplicates subjected to 10 cycles of pre-amplification. Pearson’s correlation coefficient is indicated. (**d**) Correlation of each of the technical duplicates from the same nCounter run with the results obtained in an independent nCounter assay of the same sample. Pearson’s correlation coefficient is indicated.

**Figure 4 pharmaceutics-14-02034-f004:**
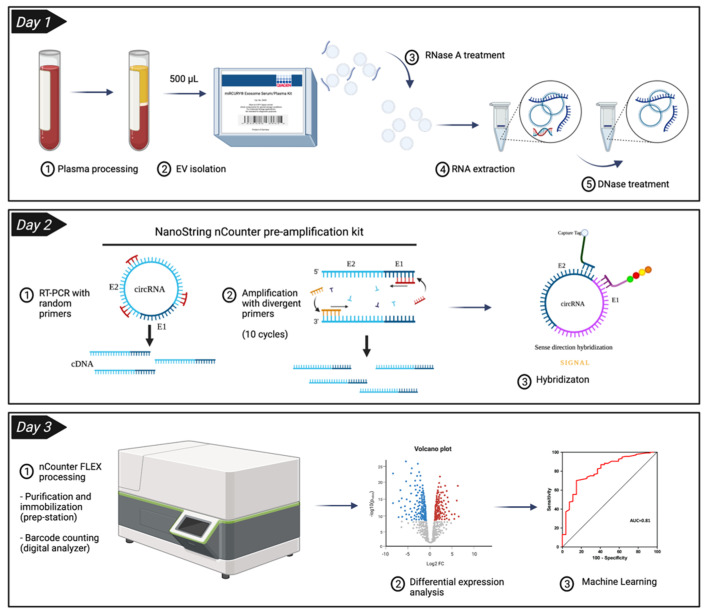
Final workflow established for the study of circRNAs from plasma extracellular vesicles (EVs) using the nCounter technology. A volume of 500 μL of plasma was used in the miRCURY kit to precipitate EVs. Rnase A was used to remove any non-vesicular RNA that could be present in the sample before proceeding with manual RNA extraction with TRI reagent. RNA samples were treated with DNase to eliminate any trace of genomic DNA, followed by retro-transcription and a pre-amplification step of 10 cycles. Finally, samples were hybridized overnight before nCounter processing.

**Figure 5 pharmaceutics-14-02034-f005:**
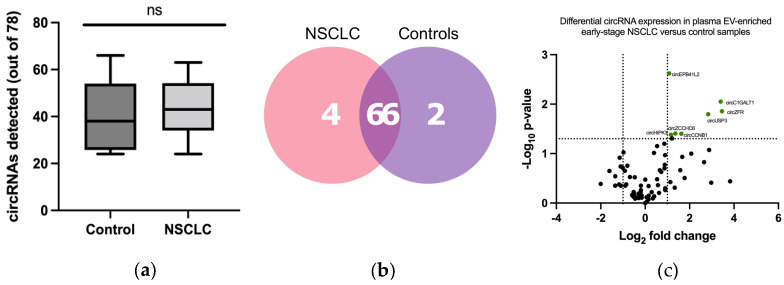
EV-circRNA detection and differential expression analysis. (**a**) Number of circRNAs detected in extracellular vesicle (EV) enriched samples from cancer patients and non-cancer controls using our custom circRNA nCounter panel, which targets 78 circRNA (Mann–Whitney U test, *p* = 0.3807); (**b**) Venn diagram displaying circRNAs identified in early-stage NSCLC and non-cancer controls, featuring those shared by both cohorts; (**c**) Differential expression analysis of log_2_-normalized counts between the early-stage NSCLC and control EV samples. circEPB41L2, circC1GALT1, circZFR, circUSP3, circZCCHC6, circHIPK3 and circCCNB1 were found upregulated in NSCLC samples.

**Figure 6 pharmaceutics-14-02034-f006:**
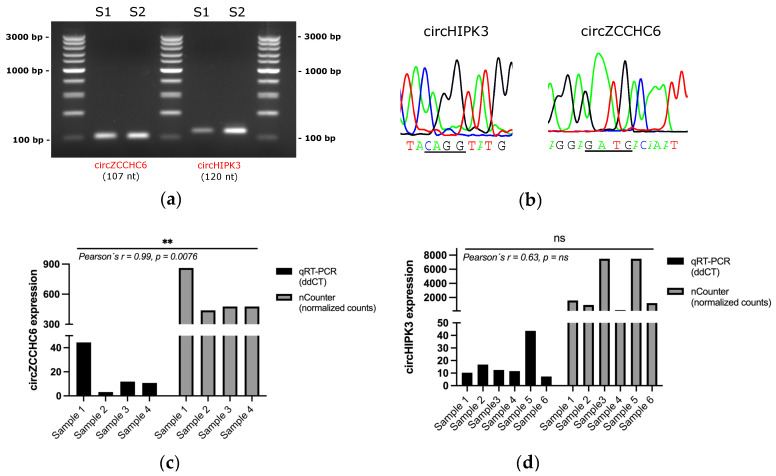
Validation of nCounter results by RT-qPCR and further Sanger sequencing. (**a**) Electrophoresis gel of amplified circZCCHC6 (107 nt) and circHIPK3 (120 nt); (**b**) Sanger sequencing results spanning the junction site (underlined) of cited circRNAs; Comparison of nCounter normalized counts versus ∆∆Cts values by RT-qPCR for circZCCHC6 (**c**) and circHIPK3 (**d**) in analyzed samples. Pearson’s correlation coefficient is indicated. ns, not significant. ** means two grades of significant (*p* < 0.01).

**Figure 7 pharmaceutics-14-02034-f007:**
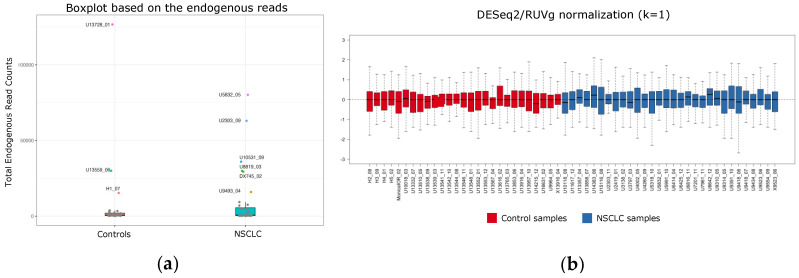
Data outlier detection and normalization for machine learning (ML) processing. (**a**) Outlier detection using the 1.5 IQR rule; (**b**) RUVSeq/DESeq2 RLE plot of normalized data (k = 1).

**Figure 8 pharmaceutics-14-02034-f008:**
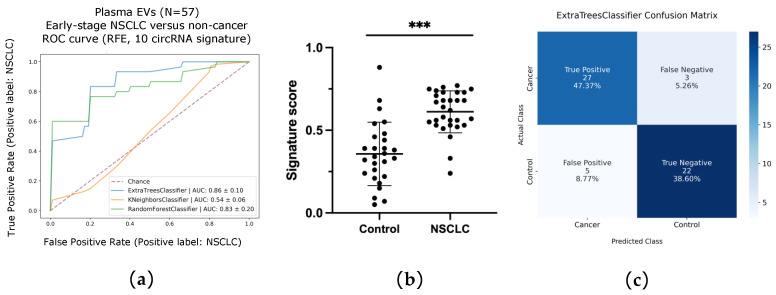
Machine learning (ML) analysis of extracellular vesicle (EV)-enriched samples. (**a**) Area under the ROC curve of the 10 circRNA-signature using recursive feature elimination (RFE) for cohort classification; (**b**) Scores of early-stage NSCLC versus control samples based on expression of the 10-circRNA signature (*p* < 0.001 in a two-tailed Mann–Whitney U test); (**c**) Confusion matrix based on the ETC classification scores. ***: *p* < 0.001.

**Figure 9 pharmaceutics-14-02034-f009:**
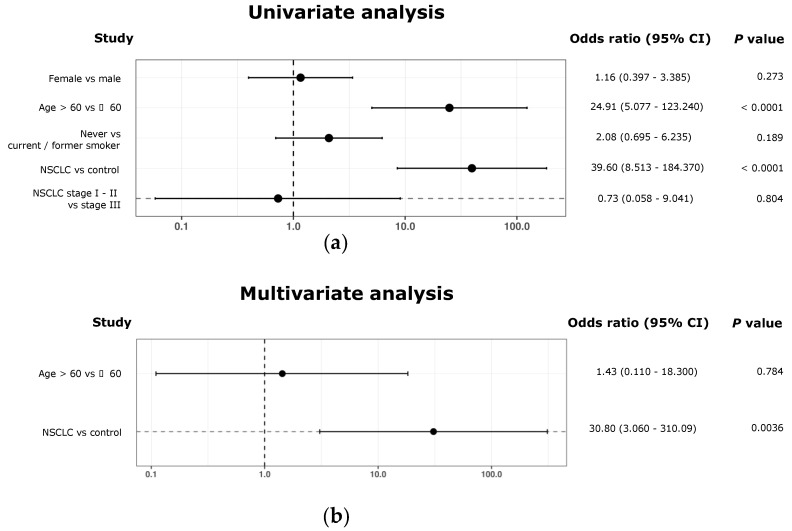
Association between clinical characteristics and ML-generated 10-circRNA signature. (**a**) Univariate analysis exploring associations between presented 10-circRNA signature and patient characteristics. Forest plot represents the odds ratios with a 95% Wald confidence limit. (**b**) Multivariate analysis exploring associations between presented 10-circRNA signature with age and cancer status. Forest plot represents the odds ratios with a 95% Wald confidence limit.

**Table 1 pharmaceutics-14-02034-t001:** Clinicopathologic characteristics of enrolled patients (*n* = 66).

Clinicopathological Characteristics	NSCLC Patients (*n* = 36)	Non-Cancer Controls (*n* = 30)
**Gender—no. (%)**		
Male	18 (50.0)	13 (43.3)
Female	18 (50.0)	17 (56.7)
**Age—yr.**		
Median	71.5	38
Range	32–91	23–57
**Histological type**		
Adenocarcinoma	27 (75.0)	-
Squamous carcinoma	4 (11.1)	-
Not information	5 (13.9)	-
**Smoking status—no. (%)**		
Former- or current smoker	20 (55.5)	11 (36.6)
Never smoker	13 (36.2)	17 (56.7)
Not information	3 (8.3)	2 (6.7)
**Tumor stage—no. (%)**		
I	19 (52.8)	-
II	2 (5.5)	-
IIIA	15 (41.7)	-

**Table 2 pharmaceutics-14-02034-t002:** Primer and probe design for circRNA validation by RT-qPCR.

CircRNA	Primers and Probes	Sequence
circHIPK3	Forward	5′CGGCCAGTCATGTATCAAAGAC 3′
	Reverse	5′AAAGGCACTTGACTGAGTTTGATAAA 3′
	Probe	FAM 5′AATCTCGGTACTACAGGTATG 3′ MGB
circZCCHC6	Forward	5′AGATGTTGTCGAATTTGTGGAAAA 3′
	Reverse	5′TCTTCTACCATTGATAAAAGCCTTCAT 3′
	Probe	FAM 5′GAGGAGAAATGACAAATT 3′ MGB

**Table 3 pharmaceutics-14-02034-t003:** Precision assessment of the ML generated circRNA signature with ETC, RF and KNN. The 95% CI are indicated.

Model	ETC	RF	KNN
No. concordant samples	49	44	30
No. discordant samples	8	13	27
AUC ROC	0.86	0.83	0.54
Accuracy	86%	77%	53%
Sensitivity	90% (CI = 73.47–97.89%)	83% (CI = 65.28–94.36%)	50% CI = 31.30–68.70%)
Specificity	81% (CI = 61.92–93.70%)	70% (CI = 49.82–86.25%)	56% CI = 41.83–68.49%)
PPV	84% (CI = 70.81–92.32%)	76% (CI = 63.10–85.10%)	56% (CI = 41.83–68.49%)
NPV	88% (CI = 71.18–95.61%)	79% (CI = 62.20–89.77%)	50% (CI = 37.95–62.02%)
Cohen’s κ	0.72 (CI = 0.458–0.976)	0.54 (CI = 0.281–0.798)	0.06 (CI =−0.202–0.313)

ML = machine learning, AUC = area under the curve, ROC = receiver operating characteristic, RF = random forest, KNN = K-nearest neighbor, CI = confidence interval, PPV = positive predictive value, NPV = negative predictive value.

**Table 4 pharmaceutics-14-02034-t004:** Association between age and cancer status.

Statistic	DF	Value	*p*-Value
Chi-Square	1	32.245	<0.0001
Likelihood Ratio Chi-Square	1	41.232	<0.0001

DF = Degrees of freedom.

**Table 5 pharmaceutics-14-02034-t005:** Analysis of maximum likelihood estimates.

Parameter	DF	Estimate	Standard Error	Wald Chi-Square	*p*-Value
Age	1	0.356	1.301	0.075	0.7840
Cancer status	1	3.427	1.178	8.462	0.0036

DF = Degrees of freedom.

## Data Availability

The data that support the findings of this study are available from the corresponding author (carlospedraz@icloud.com) upon reasonable request.

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
