# Peer review of "Multiplex Analysis of CircRNAs from Plasma Extracellular Vesicle-Enriched Samples for the Detection of Early-Stage Non-Small Cell Lung Cancer"

_pharmaceutics, 2022, doi:10.3390/pharmaceutics14102034_

Round 1

Reviewer 1 Report

This is a well-written story that has the novelty of validating nCounter platform for multiplexed EV-circRNA expression studies in lung cancer patients. The authors have shown promising results by developing protocol for determining EV-circRNA expression along with differential expression and predictive machine learning studies. While further studies are warranted to develop its clinical application, this work provides the potential of developing this approach as a diagnostic tool for detecting lung cancer.

I propose few minor suggestions for improving the quality of the manuscript, listed below:

1. Please adhere to the referencing format of Pharmaceutics journal. The in-text citations require spacing in front of them.

2. The manuscript has inconsistencies in the use of numbers and units in several section. Examples: Line 120 mentions 'two hours'; line 121 has '10 minutes'; line 122 has wrong symbol for degree celsius; line 128 has 'ml' instead of mL; line 183 has 'five microliters' and many more. I would suggest revisiting the manuscript to modify all areas using mL or uL for units, 2 h, 10 min, so on.. consistently.

3. In Section 2.3 Line 126, the miRCURY Exosome Serum/Plasma Kit was used but doesn't mention any method of how this was carried out. I would suggest writing the protocol in brief, if possible, otherwise mention 'according to manufacturer's protocol'.

4. Section 2.3 Line 139, it is not clear what 'buffer' was used. Please specify.

5. Section 2.4 Line 151, Image J software requires the version and company name and headquarters in brackets.

6. Section 2.6 Line 168, 'Merck' is missing the name of city and country.

7. Section 2.6 Line 172, I would recommend writing 'DNA-free DNA Removal Kit' instead of 'DNA-free Kit.

8. Section 2.7 line 184, the M-MLV reverse transcriptase enzyme is missing the company, city and country name. This particular section also has major similarities with previously published articles. I suggest revisiting this section and modify appropriately to reduce the similarities.

9. Section 2.11, RF is already defined in line 251, I suggest writing RF instead of full form in line 256. Also the full forms for PPV and NPV are missing in line 262.

10. For consistency, I would suggest using capital letters for Panels A, B, C... for all figures throughout manuscript as they are mentioned as Figure 1A or Figure 1B in text, or vice-versa using Figure 1a, 1b, 1c...

11. Section 3.1 Figure 1B, suitable y-axis titles can be added. Also, the y-axis values can be made consistent for miRCURY and UC NSCLC samples, and similarly for miRCURY and UC Control samples for facilitating visual comparisons of the graphs.

12. Section 3.1 line 307, I missed how the data '14.151 and 656.525' correspond to figure 2A. I hope the authors can clarify this or revise the figure appropriately if necessary.

13. Section 3.1 Figure 2, The legend to Figure 2 seems inadequate. I would suggest to elaborate by specifying samples from who and what machines were used to obtain data.

14. Section 3.1, the error values+/- are missing in text line 307, 308, 341, 342.

15. Section 3.2 line 406, 'the above-mentioned protocol' doesn't adequately specify where to look. Suggest adding 'at end of section 3.1' and/or figure 4, if appropirate.

16. Section 3.2, line 476, 477 and line 480 and 481, circEPB41L2, circC1GALT1, circZFR, circUSP3, circZCCHC6, circHIPK3 and circCCNB1 can be defined at the first mention.

17. Figure 6C, Since there are no lines to separate the panels A, B and C, I would recommend splitting Panel C to C and D for clarity and specificity, and make relevant modifications in text.

18. Section 4, line 633, I would suggest writing few lines discussing why the authors found differences in EV population from two isolation methods of ultracentrifugation approach and miRCURY kit.

Reviewer 2 Report

In the research article ”Multiplex analysis of circRNAs from plasma extracellular vesicle-enriched samples for the detection of early-stage non-small cell lung cancer” by Pedraz-Valdunciel et al., the authors study the EV-circRNAs in plasma samples from patient with lung cancer. There is no standardized protocol for detection of EV-circRNAs. They investigate different volumes of plasma, number of pre-amplification cycles and used the platform, nCounter, to detect these EV-circRNAs. The study suggests a specific protocol for optimized detection of EV-circRNAs with nCounter. In addition, comparing plasma from healthy subjects and patients with low stage lung cancer, the authors investigate the potential use of EV-circRNAs as liquid biomarkers. Using machine learning, a 10-circRNA signatures was able to discriminate between healthy subjects and patients with lung cancer.

Overall, the manuscript is well-written. The followings are some concerns and comments that the authors may want to consider.

Major concerns:

1)      Key experiments Figure 1-3 is based on a plasma sample from one patient (sometimes also a healthy subject), and the authors base their workflow on the results from presented in these figures. The experiments should be repeated with a larger number of biological replicates.

2)      Line 97-99. The authors state, that nCounter analysis of EV-circRNAs has not been investigated previously. The authors should check this statement (a fast search led me to a paper by Hansen et al. The transcriptional landscape and biomarker potential of circular RNAs in prostate cancer. Genome Medicine, 14:8, 2022).

Minor comments:

1)      In the abstract, circRNAs should be replaced with circular RNAs (circRNAs) when mentioned the first time.

2)      Line 75-76. In the introduction, please refer to studies that have investigated circRNAs for early detection.  

3)      Figure 7b. The legends are too tiny.

4)      Figure 8a. The legend of the x-axis has been cut in the top.

5)      Figure 8c. It is not possible to read the text in the matrix. Overall, this figure could benefit from larger text sizes.

6)      Figure 9. The authors should perform a multivariate analysis to test if the association between cancer status and ETC circRNAs signature is independent from age. They should also include i tumor stage to the analysis.

7)      Line 658. A comma is missing after circEPB41L2.
